# COVID-19 Quarantine Dramatically Affected Male Sexual Behavior: Is There a Possibility to Go Back to Normality?

**DOI:** 10.3390/jcm11092645

**Published:** 2022-05-08

**Authors:** Lorenzo Spirito, Michele Morelli, Roberto La Rocca, Luigi Napolitano, Claudia Collà Ruvolo, Lorenzo Romano, Angelo di Giovanni, Carmine Sciorio, Sergio Concetti, Emanuele Montanari, Francesca Tripodi, Ferdinando Fusco, Marco Capece

**Affiliations:** 1Department of Neurosciences, Reproductive Sciences, and Odontostomatology, University of Naples “Federico II”, 80131 Naples, Italy; lorenzospirito@msn.com (L.S.); roberto.larocca@gmail.com (R.L.R.); luiginap89@libero.it (L.N.); c.collaruvolo@gmail.com (C.C.R.); dr.angelodigiovanni@gmail.com (A.d.G.); ferdinando_fusco@libero.it (F.F.); drmarcocapece@gmail.com (M.C.); 2Department of Urology, Foundation IRCCS Ca’ Granda Ospedale Maggiore Policlinico, 20122 Milan, Italy; emanuele.montanari@gmail.com; 3University of Milan, 20122 Milan, Italy; 4Urology Unit, ASST Ospedale Manzoni, 23900 Lecco, Italy; lorenzo.romani@gmail.com (L.R.); carmine.sciorio@gmail.com (C.S.); 5Urology Unit, Surgical Department, Azienda Usl, Maggiore-Bellaria Hospital, 40133 Bologna, Italy; sergio.concetti@gmail.com; 6Institute of Clinical Sexology, 00198 Rome, Italy; francitrip@hotmail.com

**Keywords:** COVID-19, sexual well-being, erectile function, sexual satisfaction

## Abstract

We performed a monocentric longitudinal study on sexually active male patients, from May 2021 to October 2021, with SARS-CoV-2 infection confirmed with a nasopharyngeal reverse transcriptase polymerase chain reaction (RT-PCR). The questionnaires were delivered by email. The study period was divided into the periods before getting tested (T1), during quarantine (T2), 1 month after a negative test (T3), and 3 months after a negative test (T4). All participants were invited to complete these questionnaires: 10- and 6-item questionnaires, a sexual distress schedule (SDS), and the international index of erectile function questionnaire of 15 items (IIEF-15). The primary endpoint was to evaluate the impact of quarantine on male sexual function (SF) during and after the SARS-CoV-2 infection. A total of 22 male patients met the inclusion criteria. The differences for both SDS and IIEF-15 scores, between T1–T2 (27 (IQR 24.0–32.2) vs. 37.5 (IQR 34.2–45.5), 45 (IQR 38.0–50.2) vs. 28.5 (IQR 19.5–38.0)), T2–T3 (37.5 (IQR 34.2–45.5) vs. 28 (IQR 24.0–31.0), and 28.5 (IQR 19.5–38.0) vs. 39.5 (IQR 35.5–44.2)) were statistically significant (*p* < 0.001), respectively. Moreover, between T1–T4, no statistically significant difference (*p* > 0.05) was recorded in both SDS (27 (IQR 24.0–32.2) vs. 26.5 (IQR 24–30.2)) and IIEF-15 (45 (IQR 38.0–50.2) vs. 28.5 (IQR 19.5–38.0)). In 20 patients (90.9%), SARS-CoV-2 had a huge impact on relationship and sexual life, but no patient attended a clinic for sexual difficulties. In conclusion quarantine has negatively influenced SF in infected patients; however, 3 months after the rRT-PCR negative test, a promising return to the preinfection SF values is observed.

## 1. Introduction

Severe acute respiratory syndrome coronavirus 2 (SARS-CoV-2) has quickly spread worldwide since the Wuhan Municipal Health Commission (China) informed the World Health Organization (WHO) of a novel pneumonia of unknown etiology in December 2019 [1].

On 8 March 2020, an unprecedented self-isolation ordinance was launched by the Italian government, influencing numerous aspects of the social life of the Italian population. The literature stated that elders are more vulnerable to severe disease, and the survival rate is much higher in younger patients [2]. Nevertheless, to date more than 10 million people have contracted the disease with more than 145,000 deaths in Italy without any sign of the slowing down of the virus—with the most recent Omicron variant first identified in South Africa at the end of November 2021 [3].

Considering the exceptional situation and the effect that such measures could have on sexual health, it is fundamental to evaluate whether this has also generated an array of sexual and reproductive issues. SARS-CoV-2 has been detected in various fluids: saliva, respiratory fluids, blood, urine, feces, and semen [4,5]. Despite that, no record of sexual transmission has been reported but it has been hypothesized that COVID-19 could have an impact on fertility, perhaps via viral orchitis, testicular injury, and inflammatory infiltration [6].

At this moment, SARS-CoV-2 is under the magnifying glass of the scientific community to better explain the pathophysiology, the spread, and the potential consequences of such disease. On the other hand, few data suggest whether and how the coronavirus could affect the sexuality of these patients.

The main purpose of our study was to evaluate, during and after the quarantine period, the impact of such disease on male sexual function (SF) in SARS-CoV-2 positive patients.

## 2. Materials and Methods

### 2.1. Study Population and Data Collection

We prospectively included all male patients diagnosed with SARS-CoV-2 with a positive nasopharyngeal reverse transcriptase polymerase chain reaction (RT-PCR) test between May 2020 to October 2020. In order to be eligible, the individual had to be aged 18 or over and in a steady relationship (intended as a romantic relationship of at least 6 months with vaginal sexual intercourse) during the COVID-19 emergency. Sexual orientation did not represent a criterion of exclusion. The questionnaires were administered by email the urologist who conducted the study via email. Baseline variables were included in a dedicated database: patient age, gender, body mass index (BMI), smoking status, Charlson score, number of children, educational level, working status, working from home status, intensive care unit (ICU) admission, length of stay (LOS) in the ICU, and housemates positive to SARS-CoV-2. An institutional ethical approval was not necessary; however, we followed the ethical principle of declaration of Helsinki.

### 2.2. Sexual Function Questionnaires

Male SF involves a complex interaction of physiological and subjective processes. To assess SF in SARS-CoV-2 positive patients **,** the participants were asked to complete various questionnaires sent via email. All patients were asked to answer the 15-item international index of erectile function (IIEF-15) questionnaire [7], a sexual distress schedule (SDS) [8], and two internally made questionnaires of 10 and 6 items investigating sexual behaviors.

The IIEF-15 questionnaire [7], translated into Italian, which is divided into five categories: overall satisfaction, erectile function, orgasmic function, sexual desire, and intercourse satisfaction was administered. The answers to the questionnaire were categorized from 0 to 5, where “0” was recorded as no sexual activity, with a final score ranging from 5 to 25.

The sexual distress schedule (SDS) [8] consisted of 12 items that correlate to different features of sexual distress [8]. Every response option ranged from 0 to 4 (never (0), not often (1), occasionally (2), often (3), and always (4)), with a maximal score of 48 that is associated with a higher level of sexual distress.

The 10-items questionnaire on the impact of COVID-19 comprised four domains: sexuality, relationships, physical health, and mental health. The six-item questionnaire concerned the individuals’ and couples’ relationship statuses. It comprised how often pornography was used, the mean frequency of vaginal sexual intercourse, frequency of autoerotism, SF difficulties, and if the respondent attended a clinic for his difficulties.

Patients were followed with IIEF, SDS, and ten- and six-item questionnaires during the study period, which was divided into the periods before getting tested (T1), during quarantine (T2), 1 month after a negative test (T3), and 3 months after a negative test (T4). Only the complete questionnaires were recorded in the analysis.

The primary outcome was to measure the influence of COVID-19-related quarantine on sexual function and sexuality in SARS-CoV-2-positive male patients.

### 2.3. Data Analysis

Descriptive statistics included frequencies and proportions for categorical variables. Mean, median, and interquartile ranges (IQR) were reported for continuously coded variables. The Wilcoxon sign rank test for paired samples was used to compare continuous nonparametric variables. In all statistical analyses, R software environment for statistical computing and graphics (R version 3.6.1, The R Foundation, Indianapolis, IN, USA) was used. All tests were two-sided with a level of significance set at *p* < 0.05.

## 3. Results

### 3.1. Study Population

A total of 78 consecutive male patients were included in the study between May 2020 and October 2020. Among them, we identified 22 sexually active patients that met the inclusion criteria: that they completed the entire questionnaires and presented an infection of SARS-CoV-2, documented with a nasopharyngeal RTPCR. The baseline characteristics are presented in Table 1. In the overall cohort, the median age was 63 (IQR 58.2–67) years and all were male patients. The median BMI was 24.6 (IQR 21.7–28), and the median Charlson comorbidity score was 3 (IQR 1–4). Of all, 9 (40.9%) patients were admitted into the Intensive Care Unit (ICU) with a median length of stay (LOS) of 10 days (IQR: 8–13).

### 3.2. Ten- and Six-Item Questionnaires

During Time 1, 22 (100%) patients reported having complete vaginal sexual intercourse. Of all, 8 (36.4%) and 5 (22.7%) patients reported having sexual intercourse more than once a month and more than once a week, respectively. Twenty patients reported having sexual difficulties, but only 2 (9.1%) reported that they were attending a clinic for sexual difficulties (both patients had trouble keeping an erection) (Table 2).

During Time 2, 5 (22.7%) patients reported having complete vaginal sexual intercourse. Of those, 2 (9.1%) and 3 (13.6%) patients reported having sexual intercourse more than once a month and more than once a week, respectively. All patients reported having sexual difficulties (Table 2).

During Time 3, 16 (72.7%) patients reported having complete vaginal sexual intercourse. Of those, 4 (18.2%) and 1 (4.5%) patient reported having sexual intercourse more than once a month and more than once a week, respectively. All patients reported having sexual difficulties, but no patient attended a clinic for sexual difficulties (Table 2).

In Table 3 we showed that 8 patients (36.4%) tended to self-isolate during the COVID-19 emergency. In addition, 9 (41%) and 17 (77.3%) patients were worried at the time of questioning for their selves and their relatives, respectively. Moreover, of 22 patients, 15 (68.2%) and 14 (63.6%) patients reported an impact of quarantine on their physical and mental health, respectively.

Fifteen (68.2%) patients reported a negative impact on their relationships and 20 (90.9%) reported a negative impact on their sexuality. None of them received any advice regarding sexual intercourse during the COVID-19 emergency (Table 3).

### 3.3. Sexual Distress Schedule

From Time 1 to Time 2, total sexual distress (SD) score increased significantly (27 (IQR 24.0–32.2) vs. 37.5 (IQR 34.2–45.5), *p* < 0.001). From Time 2 to Time 3, total SD score decreased significantly (37.5 (IQR 34.2–45.5) vs. 28 (IQR 24.0–31.0), *p* < 0.001). From Time 3 to Time 4 (*p* = 0.06), from Time 1 to Time 3 (*p* = 0.8), and from Time 1 and Time 4 (*p* = 0.1), no statistically significant differences were recorded (Figure 1). The changes of specific items (SD1–12) are shown in Table 4.

### 3.4. International Index of Erectile Function Questionnaire

From Time 1 to Time 2, the total IIEF score decreased significantly (45 (IQR 38.0–50.2) vs. 28.5 (IQR 19.5–38.0), *p* < 0.001). From Time 2 to Time 3, the total IIEF score increased significantly (28.5 (IQR 19.5–38.0) vs. 39.5 (IQR 35.5–44.2), *p* < 0.001). From Time 3 to Time 4, the total IIEF score increased significantly (39.5 (IQR 35.5–44.2) vs. 42 (IQR 36.0–48.0), *p* < 0.01). From Time 1 to Time 3, the total IIEF score decreased significantly (45 (IQR 38.0–50.2) vs. 39.5 (IQR 35.5–44.2), *p* < 0.001). Finally, from Time 1 to Time 4, no statistically significant difference was recorded (*p* = 0.09) (Figure 1). The changes of specific domains (EF, OF, SD, IS, OS) are shown in Table 5.

## 4. Discussion

The study showed a decrease in sexual function and quality of life in sexually active men during the quarantine period in SARS-CoV-2-positive patients, but 3 months afterwards (T4) a promising return to the prequarantine values was observed.

Certainly, isolation put an amount of emotional distress on couples all around the world. Social distancing attempt to restrain the virus, but it also pushed us to modify or suppress our desire for intimacy, appearing to be the leading issue for couples in order to reprogram their sexual life.

The decline in sexual activity during the quarantine, as shown in our study, is closely connected with psychological and mental health and it is not surprising that in both genders sexual health status lowered during this pandemic [9,10].

Jannini et al. showed solid proofs supporting EF as an exceptional surrogate of general good health [11]. Considering the fact that 40.9% of our patients were treated in the intensive care unit, sexual dysfunction could be amplified by severe illness, physical distress, and reduced oxygen saturation that could compromise erectile function [12].

The intensified use of pornography and autoerotism during and after quarantine might have showed an unaltered level of sexual desire; nevertheless, the majority of our patients reduced the mean frequency of sexual intercourse compared to the prepandemic period. The need to satisfy themselves without exigency a partner might advise that individuals were apprehensive in terms of SARS-CoV-2 transmission during sexual intercourse.

The majority of our patients reported an increased number of sexual difficulties during the COVID-19 emergency and a negative impact of all aspects of relationship and sexuality. Miranda et al. [13] stated that previous subclinical sexual difficulties could be aggravated during this period. Moreover, sexual activity could be negatively impacted by the greater inclination to a lower level of self-care and the lack of knowledge regarding a precise quarantine ending point.

Therefore, the reduced level of social exchange with their partners could be explained by the fear of transmitting SARS-CoV-2 to loved ones and their family. In fact, most of our patients were worried about their relatives and kissing might be a great concern in couple dynamics in terms of risk transmission of SARS-CoV-2 [14]. Furthermore, we must consider the skepticism of keeping an enjoyable and safe sexual life during quarantine, and the suspicions of all men about fertility and sexual transmission of SARS-CoV-2 [4,5,15,16].

In the present study, the mean ranks of the Wilcoxon scale responses, during quarantine, showed a greater reduction in SDS and IIEF scores. These results of quarantine and early postquarantine sexual dysfunction could be explained by the fear of infecting the partner with SARS-CoV-2, suggesting that physical or mental pressure under stressful conditions might influence sexual arousal and pleasure. However, all scores improved after one month of healing and eventually returned to normality after 3 months, showing that isolation measures did not affect long-term male sexual functions. Unfortunately, all patients that suffered from sexual difficulties during the COVID-19 emergency did not attend a clinic or a physician, in fact psychological assistance could have been useful to an early sexual recovery and restoration of men’s psychological health.

The limitations of the present study should also be acknowledged. The main limitations of the study were the low number of participants that completed the entire questionnaires and presented an infection of SARS-CoV-2 and the noninclusion of a control group in the design of the study. Moreover, the absence of specific questions for LGBT individuals and despite the fact that sexual orientation did not constitute a reason for exclusion, we did not have any LGBT individual included in our study.

However, our study also has several strengths which lies on his methodology. This study is a longitudinal study in which individuals were evaluated before, during, and after the quarantine to prevent possible memory bias. Additionally, all subdomains of the international validated scales for examining sexual functions, such as IIEF and SDS, were evaluated.

## 5. Conclusions

Our study showed a substantial decrease of sexual functions during the quarantine with no or a slightly negative impact at 3 months. Potentially, the more time available might lead men to reconnect with their sexuality. Further longitudinal studies on a larger number of men are needed to confirm these results and any possible change in men’s sexuality.

## Figures and Tables

**Figure 1 jcm-11-02645-f001:**
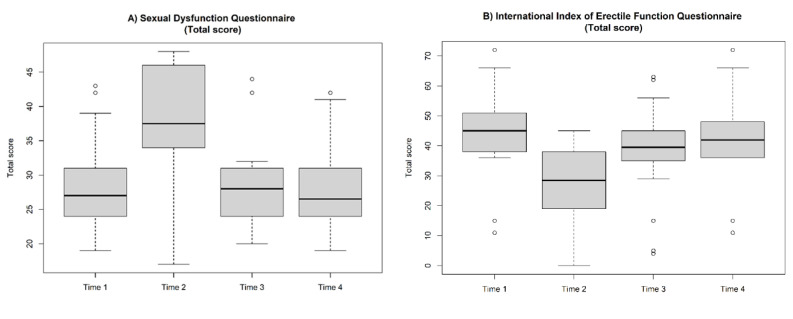
Box and whisker plots depicting sexual dysfunction (**A**) and erectile function (**B**) from the total points of the questionnaire administrated to 22 COVID-19 patients in the periods before being tested (Time 1), during quarantine (Time 2), 1 month after a negative test (Time 3), and 3 months after a negative test (Time 4). Boxes denote the interquartile range. The solid black horizontal bar denotes the median within each time period. Whiskers denote the 95% range of the distribution of tumor size. The open circles denote outlier values.

**Table 1 jcm-11-02645-t001:** Descriptive characteristics of 22 COVID-19 patients.

Overall Population		22 (100)
**Age** **(years)**	Median (IQR)	63 (58.2–67)
**Gender** **n (%)**	Male	22 (100)
	Female	0 (0)
**BMI** **(Kg/m^2^)**	<25	9 (40.9)
	25–30	11 (50.0)
	≥30	2 (9.1)
**Smoking** **(sig/day),** **n (%)**	0	10 (45.5)
	5	1 (4.5)
	10	6 (27.3)
	20	5 (22.7)
**CCS**	Median (IQR)	3 (1–4)
**N^o^ Children,** **n (%)**	0	2 (9.1)
	1	17 (77.3)
	2	3 (13.6)
**Educational level,** **n (%)**	Degree	8 (36.4)
	Primary school	4 (18.2)
	Secondary school	10 (45.5)
**Working Status,** **n (%)**	Employed	10 (45.5)
	Retired	4 (18.2)
	Self-employment	6 (27.3)
	Unemployed	2 (9.1)
**Working from home,** **n (%)**	No	21 (95.5)
	Yes	1 (4.5)
**ICU,** **n (%)**	No	13 (59.1)
	Yes	9 (40.9)
**LOS** **(days)**	Mean	11.22 (4.15)
	Median (IQR)	10 (8–13)
**Housemates** **COVID-19 positive**	No	17 (77.3)
	Yes	5 (22.7)

**Table 2 jcm-11-02645-t002:** Questionnaire administrated to 22 COVID-19 patients before COVID-19 testing (Time 1), during quarantine (Time 2), and 1 month after a negative test (Time 3).

Questions	Answers	Time 1	Time 2	Time 3
(1) Did you have complete sexual intercourse (vaginal) during this time?	No	1 (4.5)	17 (77.3)	6 (27.3)
	Yes	21 (95.5)	5 (22.7)	16 (72.7)
(2) How often did you have sexual intercourse?	Never	1 (4.5)	16 (72.7)	6 (27.3)
	Less than once per month	2 (9.1)	1 (4.5)	5 (22.7)
	Once a month	6 (27.3)	2 (9.1)	6 (27.3)
	More than once a month	8 (36.4)	3 (13.6)	4 (18.2)
	More than once a week	5 (22.7)	0 (0)	1 (4.5)
	Once a day	0 (0)	0 (0)	0 (0)
	More than one a day	0 (0)	0 (0)	0 (0)
(3) How often did you masturbate?	Never	11 (50.0)	14 (63.6)	9 (40.9)
	Less than once per month	7 (31.8)	2 (9.1)	10 (45.5)
	Once a month	4 (18.2)	0 (0)	3 (13.6)
	More than once a month	0 (0)	6 (27.3)	0 (0)
	More than once a week	0 (0)	0 (0)	0 (0)
	Once a day	0 (0)	0 (0)	0 (0)
	More than one a day	0 (0)	0 (0)	0 (0)
(4) How often did you watch porn?	Never	14 (63.6)	18 (81.8)	14 (63.6)
	Less than once per month	6 (27.3)	1 (4.5)	8 (36.4)
	Once a month	2 (9.1)	0 (0)	0 (0)
	More than once a month	0 (0)	3 (13.6)	0 (0)
	More than once a week	0 (0)	0 (0)	0 (0)
	Once a day	0 (0)	0 (0)	0 (0)
	More than one a day	0 (0)	0 (0)	0 (0)
(5) Were you having sexual difficulties?	No	0 (0)	0 (0)	0 (0)
	Yes	22 (100)	22 (100)	22 (100)
(6) If Yes, have you ever had such difficulties?	No	0 (0)	0 (0)	0 (0)
	Yes	22 (100)	22 (100)	22 (100)
(7) Did you attend a clinic for your difficulties?	No	20 (90.9)	22 (100)	22 (100)
	Yes	2 (9.1)	0 (0)	0 (0)

**Table 3 jcm-11-02645-t003:** COVID-19 positivity period.

Questions		22 (100%)
1. Could you easily self-isolate?	No	14 (63.6)
	Yes	8 (36.4)
2. Are you worried at the moment for yourself?	No	13 (59.1)
	Slightly worried	6 (27.3)
	Moderately worried	3 (13.6)
	Very worried	0 (0)
3. Are you worried at the moment for your relatives?	No	5(22.7)
	Slightly worried	4 (18.2)
	Moderately worried	12 (54.5)
	Very worried	1 (4.5)
4. Have you followed the quarantine protocols?	Completely	22 (100)
5. Did it impact on your physical health?	Not at all	1 (4.5)
	Not very much	6 (27.3)
	Very much	7 (31.8)
	Completely	8 (36.4)
6. Did it impact on your mental health?	Not at all	1 (4.5)
	Not very much	7 (31.8)
	Very much	10 (45.5)
	Completely	4 (18.2)
7. Did the COVID-19 emergency impact on your relationship?	Not at all	1 (4.5)
	Not very much	6 (27.3)
	Very much	9 (40.9)
	Completely	6 (27.3)
8. How did the COVID-19 emergency impact on all aspects of your relationship?	Made them much worse	8 (36.4)
	Made them worse	12 (54.5)
	Stayed the same	2 (9.1)
	Made them better	0 (0)
	Made them much better	0 (0)
9. How did the COVID-19 emergency impact on your sexuality?	Very badly	13 (59.1)
	Badly	7 (31.8)
	No difference	2 (9.1)
	Improved	0 (0)
	Much improved	0 (0)
10. Did you receive any information or advice regarding the sexual intercourse during COVID-19 emergency?	No	22 (100)
11. Did you receive any information or advice regarding having pregnancy during COVID-19 emergency?	No	22 (100)

**Table 4 jcm-11-02645-t004:** Sexual distress schedule administrated to 22 COVID-19 patients in the periods before being tested (Time 1), during quarantine (Time 2), 1 month after a negative test (Time 3), and 3 months after a negative test (Time 4).

						Wilcoxon Sign Rank Test *p*-Value					
		Time 1	Time 2	Time 3	Time 4	Time 1 vs. Time 2	Time 2 vs. Time 3	Time 3 vs. Time 4	Time 1 vs. Time 3	Time 1 vs. Time 4	Time 2 vs. Time 4
**SD 1**	Median	2	3	2	2	0.001	0.001	0.3	0.3	1.0	0.001
	IQR	2–2.8	2.2–4.0	2.0–2.0	2.0–2.8						
**SD 2**	Median	2	3	2	2	<0.001	0.001	0.2	0.2	1.0	<0.001
	IQR	2.0–2.8	3.0–4.0	2.0–3.0	2.0–2.8						
**SD 3**	Median	2	3	2	2	0.002	<0.001	1.0	0.3	0.3	0.001
	IQR	2.0–3.0	3.0–4.0	2.0–3.0	2.0–3.0						
**SD 4**	Median	2	3	2	2	0.001	0.001	0.2	0.5	0.3	<0.001
	IQR	2.0–3.0	3.0–4.0	2.0–3.0	2.0–2.8						
**SD 5**	Median	2	3.5	2	2	0.004	0.01	0.2	0.2	1.0	<0.01
	IQR	2.0–3.0	3.0–4.0	2.0–3.8	2.0–3.0						
**SD 6**	Median	2	3	2	2	0.005	<0.001	0.8	0.3	1.0	<0.01
	IQR	2.0–3.0	3.0–4.0	2.0–3.0	2.0–3.0						
**SD 7**	Median	2	3	2	2	0.03	0.09	0.2	0.2	1.0	0.03
	IQR	2.0–3.0	2.0–4.0	2.0–3.0	2.0–2.8						
**SD 8**	Median	2	3	2	2	<0.001	<0.001	0.2	1.0	0.1	<0.001
	IQR	2.0–2.0	3.0–4.0	2.0–2.0	2.0–2.0						
**SD 9**	Median	2.5	3	2	2.5	0.04	0.01	0.4	0.1	1.0	0.03
	IQR	2.0–3.0	3.0–4.0	2.0–3.0	2.0–3.0						
**SD 10**	Median	2	3	2	2	<0.01	0.01	0.07	0.1	1.0	<0.01
	IQR	2.0–3.0	2.2–4.0	2.0–3.0	2.0–3.0						
**SD 11**	Median	2	3	2	2	<0.001	0.01	0.01	0.03	1.0	<0.001
	IQR	2.0–2.0	3.0–4.0	2.0–3.0	2.0–2.0						
**SD 12**	Median	2	3	2	2	<0.01	<0.01	0.1	1.0	0.1	<0.001
	IQR	2.0–3.0	3.0–4.0	2.0–3.0	2.0–3.0						
**SD TOT**	Median	27	37.5	28	26.5	<0.001	<0.001	0.06	0.8	0.1	<0.001
	IQR	24–30.2	34.2–45.5	24.0–31.0	24.0–30.2						

**Table 5 jcm-11-02645-t005:** International index of erectile function (IIEF) questionnaire administrated to 22 COVID-19 patients in the periods before COVID 19 testing (Time 1), during quarantine (Time 2), 1 month after a negative test (Time 3), and 3 months after a negative test (Time 4).

						Wilcoxon Sign Rank Test *p*-Value					
		Time 1	Time 2	Time 3	Time 4	Time 1 vs. Time 2	Time 2 vs. Time 3	Time 3 vs. Time 4	Time 1 vs. Time 3	Time 1 vs. Time 4	Time 2 vs. Time 4
**IIEF-EF**	Median	15	10	13	14	<0.001	<0.001	0.07	<0.001	0.07	<0.001
	IQR	12.2–18.5	5.0–12.0	10.2–15.8	10.0–18.5						
**IIEF-OF**	Median	5	4.5	5	5	<0.001	<0.001	0.5	0.01	0.3	<0.001
	IQR	5.0–6.8	2.0.2–5	5.0–5.8	5.0–6.0						
**IIEF-SD**	Median	7	5	5	5.5	<0.001	0.3	0.01	<0.001	0.1	<0.01
	IQR	5.0–8.0	2.2–6.8	3.2–7.0	5.0–8.0						
**IIEF-IS**	Median	10	5	9	10	<0.001	0.001	0.06	0.01	1.0	<0.001
	IQR	8.0–10.8	5.0–7.8	7.2–10.0	8.0–10.0						
**IIEF-OS**	Median	7	5	7	7	<0.001	0.004	0.5	0.053	0.2	0.001
	IQR	6.0–8.0	3.0–6.8	6.0–8.0	6.0–8.0						
**IIEF-TOT**	Median	45	28.5	39.5	42	<0.001	<0.001	<0.01	<0.001	0.09	<0.001
	IQR	38.0–50.2	19.5–38.0	35.5–44.2	36.0–48.0						

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
