# Peer review of "COVID-19 Quarantine Dramatically Affected Male Sexual Behavior: Is There a Possibility to Go Back to Normality?"

_jcm, 2022, doi:10.3390/jcm11092645_

Round 1

Reviewer 1 Report

The authors considered the effect of COVID-19 quarantine on sexual behavior; however, the following questions should be clarified before publication:

Minor revisions

  1. Write the abbreviation of all words where they are mentioned for the first time within the manuscript (abstract & text body). For instance, the abbreviation of SF should be written where it has been mentioned for the first time either in the abstract or manuscript body.
  2. It was better if you wrote some of main findings or scores as quantitative or mean ±SD within the abstract.
  3. Error bars in bar and linear charts should be explained whether they are standard error bars (SE) or standard deviation (SD)?
  4. As you considered sex behaviour in male patients, you should enter “male sex” in the manuscript title.

Major revisions

  1. It would be well if the authors consider the quality of sperm and serum levels of sex hormones in each case.
  2. Why you didn’t enter another group as a control?
  3. I highly recommend evaluating sex hormones and quality of sperm between patients and controls.

Author Response

We thank the Reviewer for the pertinent comment. However, due to the non-normal distribution of the data, we decided to depict the median values with the respective interquartile range. Indeed, as previously described in the Figure's label, boxes denote the interquartile range. The solid black horizontal bar denotes the median within each Time period. Whiskers denote the 95% range of the value distribution. The open circles denote outlier values.

Reviewer 2 Report

English review needed – sometimes the verb is mssing, wrong  pas tense conjugation (eg ….), punctuation review

Line 17: the questionnaires (and not the study) were delivered by email

Line 18: T1 timing needs to be clarified. How long before getting tested? And how were the patients recruited? They were already followed in Urology apointments for other reasons? Did you deliver questionnaires to random male patients and tracked the ones who tested positive? None of this is clear

Line 22: abbreviation of “SF” was not clarified; and a definition of sexual function must be given – desire? Behavior? Performance?

Please state that only men were included – it is implicit but it should be explicitly stated (abstract, methods) – and that choice should be mentioned and explained in the discussion

Line 61 – you need to define “steady relationship” – fixed unique partner? Did you exclude casual multiple partners? That needs to be clarified

Line 65 – what do you mean by smart working?

Line 91: this needs to be better explained

Line 111: you only asked for penetration intercourse? That should be clarified, and stated if oral sex was not taken into account (it is written in the questionnaire, but not in the text)

Line 114: if “sexual difficulties” were not specified, that should also be stated, because it is a subjective analysis

Line 175 – discussion – must be revised and completed – se below

You have to discuss the results taking into account that some patients were in ICU… no sexual activity by principle… And what about the symptoms of covid? You must discuss the fact that a decrease in sexual activity and desire could simply be a result of the illness and symptoms and physical distress

Line 187 – “might have shown” instead of “showed”…. Be careful with the interpretations

Line 189-190, 197-198, 205-206 – you should cite studies regarding the impact of covid on fear and sexual behavior - http://dx.doi.org/10.1136/sextrans-2020-054834

Line 209-212 – do you think this can be implemented in such an acute period? If so, you should suggest how this could be arranged in the future

Line 218-222 – did the patients answer the questionnaires at the exact timings (meaning  during quarantine, post quarantine) or was it an active recall? – if so, than it exists a memory bias, which is not a problem but should me mentioned as a limitation. Subjectivity should also be mentioned as a limitation, partially fought with validated questionnaires

Line 223 – conclusions should be reformed in light of the above commentaries. Extrapolation statements should be used with caution. This study s is a comparision between self-reported measures in  a specific period  of time - before the infection (T1 – not specified), during and after. The value of the present study should be shown in terms of future impact.

Author Response

Dear Editor,                  

We are truly honored of the interest that the Editorial team of Journal of Clinical medicine has paid to our work. We would like to thank the reviewers for the time they spent on our paper and for their thoughtful comments and constructive suggestions, which helped us to improve the quality of this report. Please find enclosed our revised manuscript and the rebuttal letter that specifically addresses every point they raised.

Sincerely yours,

Dr Michele Morelli & Dr. Marco Capece

Round 2

Reviewer 1 Report

Dear Editor,

The authors improved the quality of manuscript accordingly. It can be accepted for publication. 

Best 

Reviewer 2 Report

none